# Distribution and Quantification of 1,2-Propylene Glycol Enantiomers in Baijiu

**DOI:** 10.3390/foods10123039

**Published:** 2021-12-07

**Authors:** Hao Xu, Yifeng Dai, Shuyi Qiu, Baoguo Sun, Xiangyong Zeng

**Affiliations:** 1Guizhou Province Key Laboratory of Fermentation Engineering and Biopharmacy, School of Liquor and Food Engineering, Guizhou University, Guiyang 550025, China; xuhao20211108@163.com (H.X.); syqiu@gzu.edu.cn (S.Q.); xyzeng1@gzu.edu.cn (X.Z.); 2Beijing Advanced Innovation Center for Food Nutrition and Human Health, Beijing Technology and Business University, Beijing 100048, China; sunbg@btbu.edu.cn

**Keywords:** 1,2-Propylene glycol, enantiomer, chiral gas chromatography, Baijiu, aroma

## Abstract

Enantiomers of 1,2-Propylene glycol (1,2-PG) were investigated in 64 commercial Chinese Baijiu including soy sauce aroma-type Baijiu (SSB), strong aroma-type Baijiu (STB), and light aroma-type Baijiu (LTB), via chiral gas chromatography (*β*-cyclodextrin). The natural enantiomeric distribution and concentration of 1,2-PG in various baijiu were studied to evaluate whether the distribution and content of the two isomers of 1,2-PG were correlated with the aroma type and storage year. The results showed that 1,2-PG has a high enantiomeric ratio and the (*S*)-configuration predominated in SSB. The average *S*/*R* enantiomeric ratio of this compound in SSB was approximately 87:13 (±3.17), with an average concentration of 52.77 (±23.70) mg/L for the (*S*)-configuration and 8.72 (±3.63) mg/L for the (*R*)-enantiomer. The (*R*)-configuration was predominant in the STB, whereas neither (*S*) nor (*R*)-form of 1,2-PG were detected in LTB. The content of the two configurations of 1,2-PG in the JSHSJ vintage of SSB showed a wave variation, with an average *S*/*R* enantiomeric ratio of 89:11 (±1.15). The concentration of (*R*)-1,2-PG in XJCTJ vintage liquors showed an upward and then downward trend with aging time, with an overall downward trend, and the concentration of (*S*)-form showed a wavy change with an overall upward trend. Except for the LZLJ-2019 vintage where both (*R*) and (*S*)-1,2-PG were present, all other samples only existed (*R*)-form, and a decreasing trend of (*R*)-enantiomer with aging time was observed. The enantiomeric ratio of 1,2-PG might be one of the potential markers for adulteration control of Baijiu as industrial 1,2-PG usually presented in the racemic mixture. Sensory analysis revealed olfactory thresholds of 4.66 mg/L and 23.92 mg/L for the (*R*)- and (*S*)-configurations in pure water respectively. GC-O showed both enantiomers exhibited different aromatic nuances.

## 1. Introduction

Chinese Baijiu is one of the six most famous distilled spirits in the world, with a history of more than 2000 years, and it plays an important role in Chinese traditional culture [1]. Baijiu is a clear and transparent fermented alcoholic beverage, usually with an ethanol content of 38−65% by volume and containing many trace flavor substances. The unique flavor components in Chinese Baijiu are produced from many resources, such as materials, fermentation, distillation, and the aging process, and which will determine its aroma style [2,3]. There are 12 aroma types of Baijiu and more than 2400 flavor compounds found in different Baijiu samples, some of which are chiral flavor substances [4,5].

Chirality is one of the important properties of nature. Different isomers of chiral compounds may exhibit different properties, including aroma property and intensity [6,7,8]. For example, the four stereoisomers of 2-methyl-tetrahydrofuran-3-thiol acetate were found to present perceptible differences in aroma characteristics and intensities in our previous work [9]. The *R*-configuration of 3-mercapto-1-hexanol in wine has a grapefruit aroma with an olfactory threshold of 50 ng/L, while the *S* configuration has a passion fruit aroma with an olfactory threshold of 60 ng/L [10,11]. In recent years, there has been an increasing demand and emphasis on identifying enantiomers in the food industry, especially for flavors, fragrances and alcoholic beverages [12]. The knowledge of the natural enantiomeric distribution and concentration of chiral compounds in alcoholic beverages can be a valuable way to assess food quality, authenticity, safety and evaluate processing and storage time effects, as well as confirm the geographical origin of product matrices, etc. [13,14,15,16,17]. Langen et al. studied the natural concentration and enantiomeric distribution of 1,2-PG in various wines to evaluate its advantages as a potential sign of wine aroma adulteration [18].

As far as we know, more than 40 chiral compounds have been investigated in alcoholic beverages, including five alcohols, such as 2,3-butanediol [19,20], 2-methyl-1-butanol [21,22], 1-phenylethanol [23], 2-butanol [19], and 1,2-PG [18]. The 1,2-PG has an asymmetric chiral carbon atom with two different enantiomers (as in Figure 1). 1,2-PG is a colorless viscous liquid with good chemical stability and various applications. It is a polyol compound with certain hygroscopicity and is widely used as solvent, antifreeze, deicing agent, and humidifier in the food industry [24,25]. The addition of 1,2-PG to foods can retain a certain amount of moisture, meanwhile it does not increase the water activity, so it is often used as a lubricant to improve the sensory qualities of food [26]. 1,2-PG was found with a high level of (*R*)-form, in beer [27], wine [18,28], sparkling wine, at the concentration level of mg/L. 1,2-PG has been applied as a potential signature compound for authenticity verification in alcoholic beverages such as wine and beer [18,27]. Understanding the natural enantiomeric composition of 1,2-PG is important for the authenticity assessing in wine, so as in Baijiu. 1,2-PG was detected in raw liquor of Maotai flavour Baijiu [29] as well as the mechanized brewing of strong-flavor Baijiu [30].

The distribution and concentration of the two isomers of 1,2-PG in Baijiu are not clear. Therefore, the aim of this work was to isolate and determine (*S*)-1,2-PG and (*R*)-1,2-PG in SSB, STB, LTB and the corresponding vintages to explore whether the distribution and content of the two isomers of 1,2-PG are correlated with the aroma type and aging year.

## 2. Materials and Methods

### 2.1. Chemicals

Anhydrous ethanol (chromatographic grade, 99.97%), Tianjin kemiou Chemical Reagent Co., Ltd., Tianjin, China. anhydrous sodium sulfate (99%) (Chengdu Jinshan Chemical Reagent Co., Chengdu, China). microfiltered water, SZ-93A Pure Water Distiller, (Shanghai Yarong Co, Shanghai, China). The chiral standards are as follows: (*R*)-1,2-PG, 98%, (Shanghai Maclean Technology Co, Shanghai, China). (*S*)-1,2-PG, 99%, (±)-1,2-PG (racemic mixture, 50:50), 99%, (TCI Shanghai Chemical Industry Development Co, Shanghai, China); 2-octanol, 98%, Dr. Ehrensorfer, Germany. The standards were stored at 4 °C in a refrigerator.

### 2.2. Samples

The determination of (*S*)- and (*R*)-1,2-PG were carried out in 38 types of SSB, 16 of STB and 10 LTB. In this paper, the 64 samples were divided into two categories, in which SSB included commercial Baijiu products, JSHSJ vintage Baijiu (five–41 years), XJCTJ vintage Baijiu (one–11 years); STB included commercially available Baijiu products and LZLJ vintage Baijiu (two–nine years); and LTB included commercially available Baijiu products.

### 2.3. Sample Preparation

A 5 mL of the experimental Baijiu sample was added directly into a centrifuge tube, and 1.86 g of anhydrous sodium sulfate was added, overnight, and then filtered through a 0.22 μm organic system filter membrane. 10 μL of the internal standard solution (2-octanol, 5 mg/L) was added to 990 μL of the Baijiu sample, filtered through a 0.22 μm organic filter membrane into a 1.5 mL injection vial, and awaited GC analysis.

### 2.4. Qualitation and Quantification of 1,2-PG Enantiomers

The two enantiomers of 1,2-PG were determined using a Thermo Fisher gas chromatograph (TRACE 1300), and the treated Baijiu sample was injected into the injection port in a split mode (injection port temperature 250 °C, split ratio: 20:1). The column was CYCLOSIL-B (30 m × 0.25 mm × 0.25 µm, (Agilent Technologies Ltd., Santa Clara, CA, USA). The oven temperature was programmed at 50 °C for 1 min, then increased at a rate of 3 °C/min to 80 °C, and finally raised by 8 °C/min to a final isotherm at 170 °C, and maintained for 5 min. Qualitative analysis: Determination of the configuration of (*S*)- and (*R*)-1,2-PG in Baijiu: first, we compared the retention times with the standards of the two enantiomers to determine the elution sequences. Next, we added the (*S*)- and (*R*)-1,2-PG standards to the Baijiu samples, respectively, and observed the changes of the corresponding peak areas of the two conformations to determine the two peaks of the two conformations in the Baijiu samples. The quantification was performed using the calibration curve established in Baijiu. All experiments were repeated three times.

### 2.5. GC-O Analysis and Evaluation

The reference compounds were sniffed by GC-O by a team of trained researchers. The olfactory analysis was performed using a Thermo Fisher gas chromatograph (TRACE 1300), coupled with a sniffer (Brechbuhler AG, Switzerland) (SNIFFER 9100) equipped with a FID and a sniffing port connected to the column outlet via a splitter. GC effluent was combined with humidified N_2_ at the bottom of the glass-sniffing nose to avoid nasal dehydration. Each enantiomer standard of 1,2-PG was injected with 1 μL, injection temperature (250 °C in split mode, split ratio: 20:1). The column was a CYCLOSIL-B (30 m × 0.25 mm × 0.25 µm, Agilent Technologies Ltd., Santa Clara, CA, USA). The oven temperature was programmed at 50 °C for 1 min, then increased at a rate of 3 °C/min to 80 °C, and finally raised by 8 °C/min to a final isotherm at 170 °C, and maintained for 5 min. The carrier gas was high purity nitrogen with a column head pressure of 50 kpa.

### 2.6. Aroma Thresholds

Ten trained evaluators aged 22–29 (five males and five females) were selected for this experiment. Olfactory detection thresholds of (*R*)- and (*S*)-1,2-PG were determined by sensory panel in microfiltered water, referring to the GB/T 22366-2008 sensory analysis methodology adopted (general guidelines for the determination of olfactory and flavor perception thresholds by the three-point option method (3-AFC)) and GB/T33406-2016 (Baijiu flavor GB/T33406-2016 (guidelines for determination of flavor thresholds of Baijiu) [31,32]. 3-AFC method: We rovided each panelist with six sets of samples, one reference sample and one known blank sample. Each set included two blanks and test compound solutions of known concentration. Each evaluation sample (including test compound solution of known concentration and blank sample) was marked with a random code of three to four digits. All the samples were poured into the clean tulip-shaped Baijiu glasses, then judged by the assessors at 20 °C ± 5 °C. We compared the evaluation result of the evaluator with the actual type of the sample to determine whether the concentration of each flavor substance was answered correctly or not, and then calculated based on the statistical results. All experiments were carried out three times. The individual best estimate threshold (BET) of each assessor was calculated based on the geometric mean of the highest concentration sample that answered incorrectly and the adjacent higher-level concentration sample. $T_{BETi}$ is the individual best estimate threshold; $A_{x}$ is the highest concentration sample that the evaluator answered incorrectly; $A_{x+1}$ is the Concentration of higher-level samples; $T_{BET}$ is the Group threshold.
$$T_{BETi}=\sqrt{A_{x}\times A_{x+1}}$$
$$T_{BET}=\sqrt[{n}]{T_{BTE1}\times T_{BET2}\ldots \times T_{BETn}}$$

### 2.7. Statistical Analysis

Data were organized according to the Microsoft Office Excel 2018 application, box line graphs, scatter plots, and histograms were produced by origin 2018 64Bit. An SPSS software (IBM SPSS Statistics 26.lnk) *t*-test was used to assess the significant differences in 1,2-PG levels in different aromatic Baijiu. The level of statistical significance was 5%, *p* < 0.05. The results of the sensory tests were statistically calculated according to the national standard specifications.

## 3. Results and Discussion

### 3.1. Enantiomeric Distribution and Concentration of 1,2-PG

The enantiomers of 1,2-PG were separated by chiral gas chromatography a with *β*-cyclodextrin phase. Figure 2 is the chromatogram of representative Baijiu samples of three different aromas: (a) is one of the Chromatogram of SSB; (b) is one of the chromatogram of STB; and (c) is one of the chromatogram of LTB. As shown in Figure 2, 1,2-PG enantiomers exhibited a good separation and (*S*)-1,2-PG was first eluted.

To further explore the enantiomers of 1,2-PG in Chinese Baijiu, qualitative and quantitative analysis was performed using GC-O. For both substances, good linearity was obtained in the ranges of 0.25–128 mg/L and 0.25–1024 mg/L for the (*R*)- and (*S*)-configurations, respectively, with R^2^ values of 0.9996 and 0.9988 for the standard curves. Furthermore, the detection limits of the analytical methods were 0.791–1.471 mg/L. The recoveries of (*R*)-1,2-PG chiral isomer standards ranged from 92.40% to 98.80% with the relative standard deviations (RSDs) of 0.81–1.54%, and the recoveries of (*S*)-1,2-PG standards ranged from 98.40% to 100.85%, with the RSDs of 1.09–5.06% (Table 1).

The enantiomeric distributions of 1,2-PG in 64 Baijiu of different aromas and vintages showed differences. The concentration levels of 1,2-PG (*S* and *R*) in SSB were higher than those of 1,2-PG in STB (*p* < 0.01). 1,2-PG enantiomers were not detected in LTB. In conclusion, the large variation in 1,2-PG content in different aromatic Baijiu suggested that the levels of 1,2-PG precursors may vary in different aromatic Baijiu. In addition, Langen et al. [18] quantified the 1,2-PG enantiomers in wines, where 1,2-PG has a high enantiomeric ratio and the (*R*)-configuration was dominant, with an *R*/*S* ratio of about 90:10. Interestingly, the enantiomeric distribution of 1,2-PG in SSB is the opposite of that reported by Langen et al. in wine. The (*S*)-configuration was dominant in SSB, with an average ratio of *S*/*R* of 87:13 (±3.17) in the SSB while in the vintage ones about 89:11 (±1.15) and 89:11 (±3.82), respectively (Table 2). The particular enantiomeric distribution of 1,2-PG with the great predominance of (*S*)-enatiomer in SSB could also be a potential marker for the adulteration.

The highest 1,2-PG content was found in SSB, with (*S*)-1,2-PG content from 22.33 (±1.75) to 101.88 (±2.72) mg/L, while it was from 0 to 15.96 (±0.91) mg/L in STB; the (*R*)-1,2-PG content ranged between 2.74 (±0.06) and 19.18 (±2.75) mg/L in SSB, and 0~6.41 (±2.23) mg/L in STB (Table 3). The overall comparison of the two enantiomers of 1,2-PG in two aromatic types (SSB and STB) of Baijiu revealed significant differences between the enantiomers, with significantly higher concentrations of 1,2-PG in SSB than in STB (*p* < 0.01).

The production of SSB is based on the production of Daqu (ferments), followed by seven repeated fermentations [33,34]. The concentrations of (*S*)- and (*R*)-1,2-PG in SSB were significantly higher than those of in STB and LTB, which may be related to the Daqu production and fermentation temperature of SSB. The concentrations and enantiomeric ratios of (*S*)-and (*R*)-1,2-PG in the SSB products were significantly different (*p* < 0.01) (Figure 3a,b). The (*S*)-configuration was predominant, with an average concentration around 52.77 mg/L and an average enantiomeric percentage around 87%. The (*R*)-configuration concentration was low, with an average concentration about 8.72 mg/L and an average enantiomeric percentage of about 13%. The concentration of (*R*)-1,2-PG in JSHSJ vintage showed a wave change with aging time and an overall decreasing trend, while the concentration of (*S*)-1,2-PG showed an increasing and then decreasing trend with aging time and an overall decreasing trend. The concentration of (*R*)-1,2-PG in the XJCTJ vintage was increasing and then decreasing with time, with an overall decreasing trend, and the concentration of (*S*)-1,2-PG was a wavy variation with an overall increasing trend (Figure 3c,d).

Unlike SSB, STB is made with the Daqu at a medium temperature, followed by alcoholic fermentation at 32–35 °C [2]. The STB is mainly distributed in the Sichuan Basin (SCB)where is characterized by warmth and high humidity throughout the year [35]. Differences in the concentrations of (*S*)-and (*R*)-1,2-PG were found in the studied STB (Figure 4a) (*p* > 0.05). (*R*)-1,2-PG was dominant, with only (*S*)-1,2-PG in some of the Baijiu samples, such as LZLJTQ and WLY, and LZLJ-2019. 1,2-PG enantiomers were not detected in some samples. Moreover, among the STB vintage Baijiu, all samples had only (*R*)-1,2-PG except for the LZLJ-2019 vintage, which had both (*R*)-and (*S*)-1,2-PG. Meanwhile, a decreasing trend of (*R*)-1,2-PG with aging time was observed in the STB vintage Baijiu (Figure 4b), probably due to the involvement of alcohols in the esterification reaction.

Compared to SSB and STB, LTB is produced mainly by low-temperature Daqu, and lower alcoholic fermentation temperatures [2]. LTB’s representative brands, such as FJ, and HXEGT, have a pure and mild flavor with mellow sweetness and a fresh aftertaste [33]. Interestingly, none of the 1,2-PG enantiomers was detected in the studied LTB.

In order to better understand the variation of (*R*)-1,2-PG and (*S*)-1,2-PG content in the different aroma-types of Baijiu, the results of the study was processed as a heat map, and the color (from blue to red) indicated the variation of the relative intensity from low to high. In addition, a dendrogram of the relationship between the enantiomeric contents of 1,2-PG in different types of Baijiu was also plotted, as shown in (Figure 4c). Cluster analysis also showed a trend: 1,2-PG was somewhat differentiated among the different aromatic Baijiu. The results indicated that the two 1,2-PG enantiomers had some differences in different aromatic Baijiu.

These results highlighted the complexity of the sources of the two enantiomers, which might come from different biosynthetic and chemical pathways. The production of liquor is processed from traditional natural fermentation which involves hundreds of microbial communities, and there are certain differences in the microbial communities of different flavors of Baijiu [36]. 1,2-PG, a flavor substance in Baijiu, has been found to be metabolized by certain microorganisms. As a natural product, the accumulation of 1,2-PG was reported in the cultivation of Clostridium thermobutylicum [37]. Suzuki and Onishi et al. found that many different genera and species of yeast could convert L-rhamnose to 1,2-PG under aerobic conditions [38]. Schütz and Radler found that Lactobacillus shortum could also produce 1,2-PG [39]. However, the origin of 1,2-PG is not clear, each enantiomer has a different pathway, and the elucidation of their metabolic pathways requires specific research. The enantiomeric distinction allows for a more accurate evaluation of fragrance and aromatic components. Furthermore, each enantiomer of many chiral compounds is known to lead to different sensory responses in consumers [20,23].

### 3.2. Odor Characteristics of 1,2-PG

The 1,2-PG enantiomers standards used in this study were of high purity (98%, 99%) and no odorous impurities were detected by the 10 judges who performed the analysis by GC-O. The (*R*)-1,2-PG was described as having a faint alcoholic, fruity, sweet aroma; while the (*S*)-configuration was characterized by an aromas of wood and alcohol.

The detection threshold of (*S*)-1,2-PG in pure water was determined by the sensory panel to be 23.92 mg/L, about five times that of the (*R*)-form (Table 4). These results showed that the stereochemistry of the molecules has a certain influence on their perception, which has been confirmed in several studies [21,40,41].

## 4. Conclusions

The natural enantiomeric distribution and concentration of 1,2-PG enriched the current knowledge of 1,2-PG enantiomers in Baijiu. The content of 1,2-PG depended on the aroma style and age of the liquor to varying degrees, and its content is in the decreasing order of SSB, STB, and LTB. The concentration of the two configurations of 1,2-PG in the JSHSJ vintage of SSB showed a wave variation and an overall downward trend, with an average *S*/*R* enantiomeric ratio of 89:11 (±1.15); the content of (*R*)-1,2-PG in the XJCTJ vintage Baijiu generally tended to decrease, while the concentration of (*S*)-1,2-PG generally tended to increase with time, with an average *S*/*R* enantiomer ratio in XJCTJ vintage Baijiu of 89:11 (±3.82). In addition, the concentration of (*R*)-1,2-PG in STB vintage Baijiu decreased with time. In particular, the enantiomeric ratios showed a predominance of the (*S*)-configuration in SSB, while the (*R)*-configuration dominated in STB. Sensory analysis showed that the olfactory threshold of (*S*)-1,2-PG was about five times that of (*R*)-1,2-PG (4.66 mg/L). The two configurations have different odors. The enantiomeric ratio of 1,2-PG might be a potential marker for adulteration of Baijiu, as the industrial 1,2-PG is usually present in racemic form. Besides, it could also be a potential marker to differentiate the aroma styles of Baijiu based on its different enantiomeric ratios. Therefore, the enantiomeric ratio of 1,2-PG may be a new way to identify the quality and the aroma styles of Baijiu. The results of this study will further facilitate quality control of Chinese Baijiu.

## 5. Patents

There is a patent which we are preparing resulting from the work reported in this manuscript.

## Figures and Tables

**Figure 1 foods-10-03039-f001:**
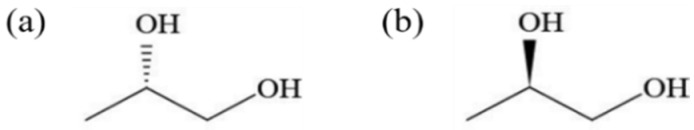
(**a**) (*S*)-1,2-PG (CAS No. 4254-15-3); (**b**) (*R*)-1,2-PG (CAS No. 4254-14-2).

**Figure 2 foods-10-03039-f002:**
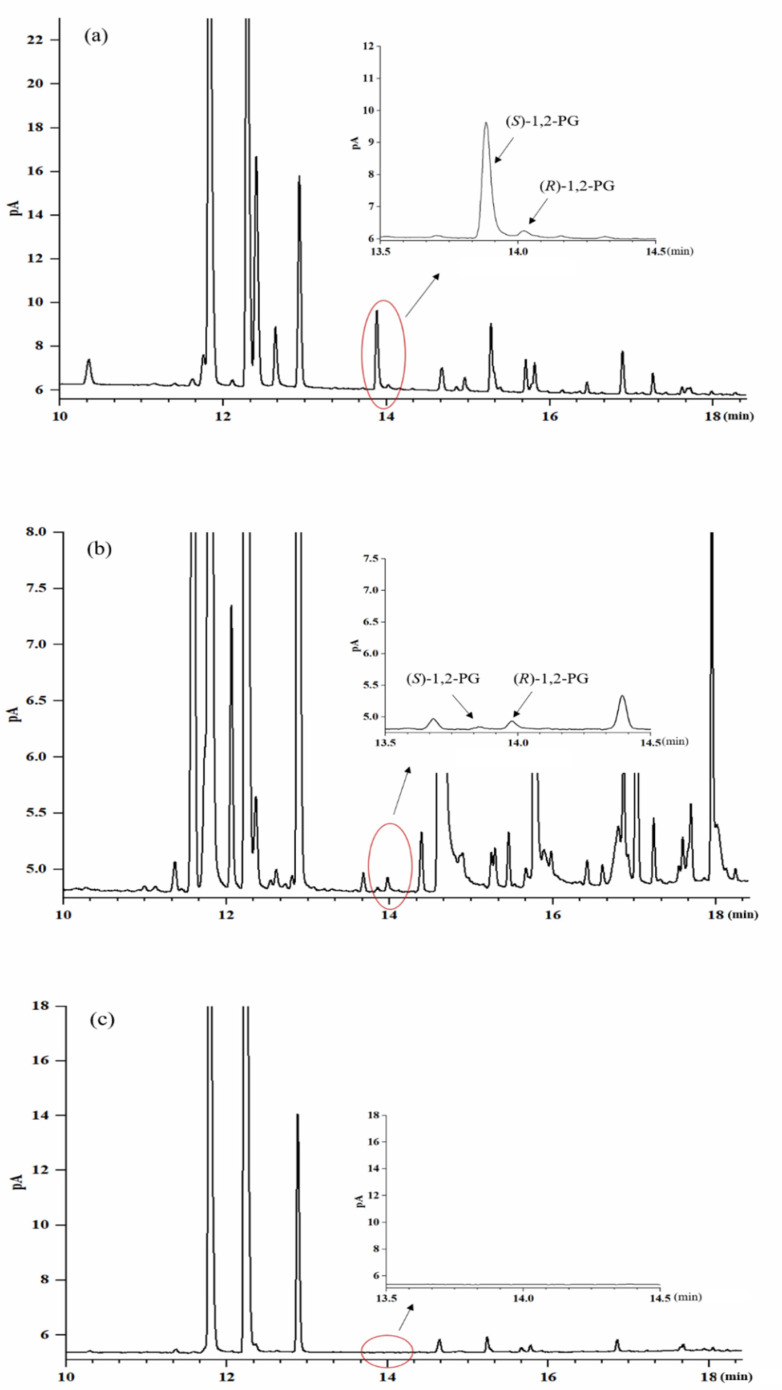
Separation chromatogram of 1,2-PG (1,2-Propylene Glycol) enantiomers in the representative Baijiu samples: (**a**) is one of SSB (soy sauce aroma-type Baijiu); (**b**) is one of STB (strong aroma-type Baijiu); (**c**) is one of LTB (light aroma-type Baijiu).

**Figure 3 foods-10-03039-f003:**
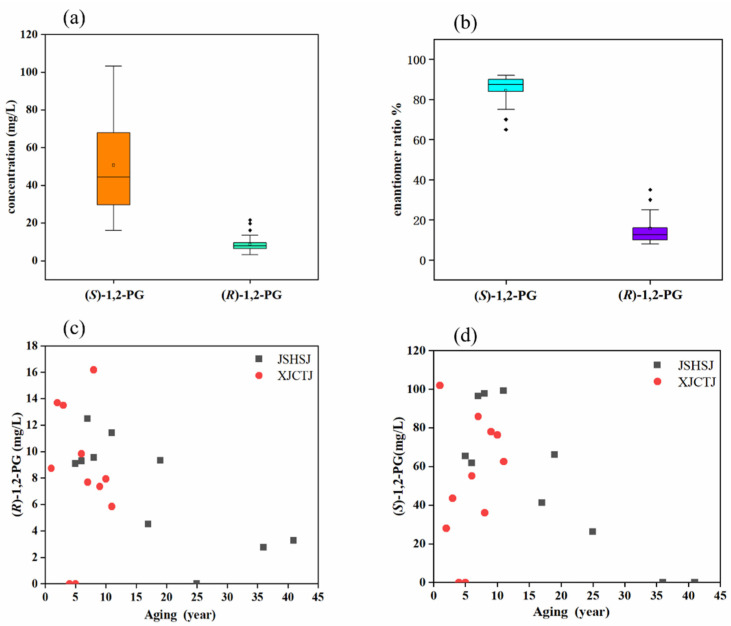
(**a**) Box line plot of 1,2-PG enantiomer content in SSB; (**b**) box line plot of 1,2-PG enantiomer ratio in SSB; (**c**) variation of (*R*)-1,2-PG content in SSB vintage Baijiu; (**d**) variation of (*S*)-1,2-PG content in SSB vintage Baijiu.

**Figure 4 foods-10-03039-f004:**
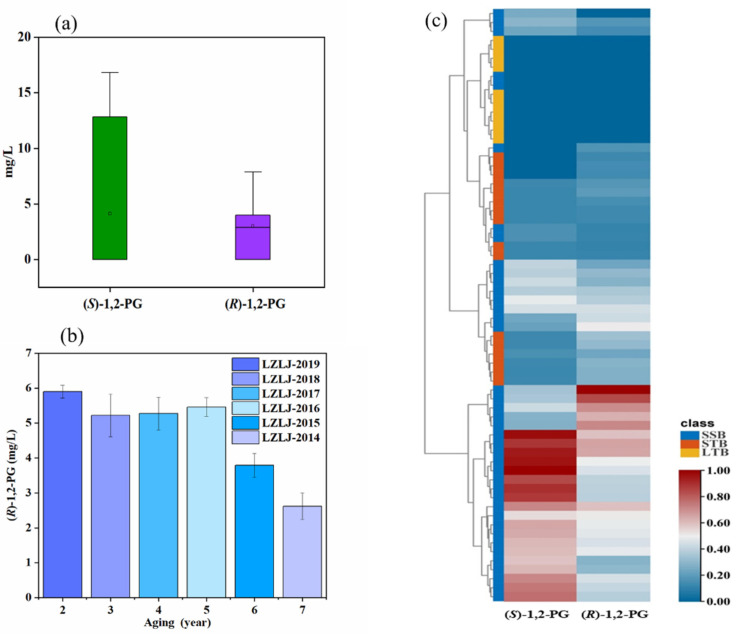
(**a**) Box line plot of 1,2-PG enantiomer content in STB; (**b**) histogram of vintage (*R*)-1,2-PG enantiomer content with aging year in STB; (**c**) thermogram analysis of 1,2-PG enantiomers in Baijiu.

**Table 1 foods-10-03039-t001:** Linear Ranges, Recovery Rates, R^2^, LOD, RSD.

No.	Compounds	Linearity (mg/L)	R^2^	RSD (%)	Recovery Rate (%)	LOD (mg/L)
1	(*R*)-1,2-PG	0.25–128	0.9995	0.81–1.54%	92.40–98.80%	1.471
2	(*S*)-1,2-PG	0.25–1024	0.9988	1.09–5.06%	98.40–100.85%	0.791

Note: R^2^: Correlation Coefficients, LOD: Limit of detection, RSD: relative standard deviation.

**Table 2 foods-10-03039-t002:** Average concentrations and ratios of 1,2-PG enantiomers in SSB.

		Mean Concentration ± Standard Deviation (mg/L) *		
Category	Number of samples	*S*	*R*	*S*/*R*
Baijiu products	17	52.77 ± 23.70 ^a^	8.72 ± 3.63 ^a^	87:13 ± 3.17
JSHSJ Vintage Baijiu (1980–2016)	10	64.15 ± 29.63 ^a^	8.54 ± 3.30 ^a^	89:11 ± 1.15
XJCTJ Vintage Baijiu (2010–2020)	11	63.01 ± 24.56 ^a^	10.08 ± 3.53 ^a^	89:11 ± 3.82

Note: *: Means in the same column with the same letter are not significantly different from each other (*p* > 0.05). SSB: soy sauce aroma-type Baijiu.

**Table 3 foods-10-03039-t003:** Enantiomeric mean concentration * ± standard deviation and ratio of 1,2-PG (1,2-Propylene Glycol) in different aromatic Baijiu.

Sample	(*S*)-1,2-PG (mg/L)	(*R*)-1,2-PG (mg/L)	ee	*S*:*R*
SSB (soy sauce aroma-type Baijiu) LM	34.91 ± 1.16 ^d^	19.18 ± 2.75 ^a^	29.09%	65:35
JSHS	90.71 ± 2.73 ^a^	12.61 ± 0.92 ^cc^	75.59%	88:12
JSJ	29.41 ± 3.95 ^ee^	3.40 ± 0.42 ^cc^	79.28%	90:10
ZJ	73.15 ± 11.05 ^bb^	11.40 ± 2.36 ^cc^	73.03%	87:13
DYT2021	62.38 ± 3.29 ^cc^	8.66 ± 0.35 ^cc^	75.62%	88:12
GT	73.04 ± 4.93 ^b^	8.55 ± 0.53 ^cc^	79.04%	90:10
XJYZ	24.57 ± 1.65 ^ee^	8.24 ± 1.15 ^cc^	49.78%	75:25
QHL	49.57 ± 6.30 ^dd^	7.30 ± 0.61 ^cc^	74.32%	87:13
LJ	91.81 ± 10.34 ^aa^	7.54 ± 0.42 ^cc^	84.82%	92:8
XJ1988	38.61 ± 0.35 ^dd^	6.89 ± 0.36 ^cc^	69.71%	85:15
TCSP	22.33 ± 1.75 ^e^	9.74 ± 1.62 ^cc^	39.25%	70:30
GZJS	39.44 ± 2.44 ^dd^	5.85 ± 0.84 ^cc^	74.15%	87:13
MTWZ	45.32 ± 1.81 ^dd^	8.53 ± 0.38 ^cc^	68.32%	84:16
DYT	89.54 ± 0.48 ^aa^	7.47 ± 0.05 ^cc^	84.59%	92:8
QJ1H	43.60 ± 1.16 ^dd^	5.40 ± 0.84 ^cc^	77.96%	89:11
MTCX	28.64 ± 0.92 ^ee^	12.07 ± 0.57 ^cc^	40.72%	70:30
MT43	60.02 ± 1.04 ^cc^	5.47 ± 0.50 ^cc^	83.29%	92:8
JSHSJ-1980	—	3.26 ± 0.14 ^cc^	—	—
JSHS-1985	23.60 ± 0.16 ^ee^	2.74 ± 0.06 ^c^	79.21%	90:10
JSHSJ-1996	26.17 ± 0.60 ^ee^	—	—	—
JSHSJ-2002	66.04 ± 3.12 ^cc^	9.31 ± 2.01 ^cc^	75.30%	88:12
JSHSJ-2004	41.15 ± 2.09 ^dd^	4.51 ± 0.24 ^cc^	80.25%	90:10
JSHSJ-2010	99.24 ± 4.73 ^aa^	11.40 ± 1.29 ^cc^	79.39%	90:10
JSHSJ-2013	97.70 ± 1.82 ^aa^	9.54 ± 2.20 ^cc^	82.20%	91:9
JSHSJ-2014	96.37 ± 4.27 ^aa^	12.49 ± 0.89 ^cc^	77.05%	89:11
JSHSJ-2015	61.72 ± 6.43 ^cc^	9.28 ± 1.48 ^cc^	73.87%	87:13
JSHSJ-2016	65.38 ± 1.28 ^cc^	9.08 ± 2.06 ^cc^	75.60%	88:12
XJCTJ-2010	62.53 ± 0.69 ^cc^	5.83 ± 0.48 ^cc^	82.96%	91:9
XJCTJ-2011	76.27 ± 1.86 ^bb^	7.93 ± 0.30 ^cc^	81.17%	91:9
XJCTJ-2012	77.93 ± 1.16 ^bb^	7.36 ± 0.02 ^cc^	82.74%	91:9
XJCTJ-2013	36.01 ± 0.28 ^dd^	16.18 ± 0.52 ^b^	37.99%	69:31
XJCTJ-2014	85.83 ± 2.63 ^a^	7.67 ± 0.29 ^cc^	83.60%	92:8
XJCTJ-2015	55.07 ± 2.14 ^c^	9.83 ± 0.33 ^cc^	69.71%	85:15
XJCTJ-2016	—	—	—	—
XJCTJ-2017	—	—	—	—
XJCTJ-2018	43.50 ± 2.79 ^dd^	13.50 ± 0.15 ^cc^	52.64%	76:24
XJCTJ-2019	28.05 ± 0.48 ^ee^	13.69 ± 0.66 ^cc^	34.39%	67:33
XJCTJ-2020	101.88 ± 2.72 ^a^	8.74 ± 0.50 ^cc^	84.20%	92:8
STB (strong aroma-type Baijiu)				
LZLJ-TOUQ	12.95 ± 0.11 ^a^	3.45 ± 0.14 ^a^	57.93%	79:21
LZLJ-TEQ	—	2.86 ± 0.08 ^a^	—	—
LZLJEQ	—	—	—	—
MZDQ	—	—	—	—
JNC	—	6.41 ± 2.23 ^a^	—	—
WLY	15.96 ± 0.91 ^a^	4.55 ± 0.57 ^a^	55.65%	78:22
GJ1573	—	2.50 ± 0.07 ^a^	—	—
SJF	—	—	—	—
LZLJ-2012	—	—	—	—
LZLJ-2013	—	—	—	—
LZLJ-2014	—	2.62 ± 0.38 ^a^	—	—
LZLJ-2015	—	3.79 ± 0.34 ^a^	—	—
LZLJ-2016	—	5.46 ± 0.27 ^a^	—	—
LZLJ-2017	—	5.27 ± 0.47 ^a^	—	—
LZLJ-2018	—	5.22 ± 0.61 ^a^	—	—
LZLJ-2019	13.09 ± 0.22 ^a^	5.90 ± 0.18 ^a^	37.85%	69:31
LTB (light aroma-type Baijiu)				
LBFJ	—	—	—	—
FJQH20	—	—	—	—
FJQXMR	—	—	—	—
FJBF	—	—	—	—
JXB	—	—	—	—
YTX1988	—	—	—	—
FPLJ	—	—	—	—
HXEGT	—	—	—	—
NLSCN	—	—	—	—
NLSEGT	—	—	—	—

Note: —: Not detected in the sample; *: Means in the same column with the same letter are not significantly different from each other (*p* > 0.05), the different letters indicate that the differences between the groups are significant (*p* < 0.05). Different letters (a, b, c, d, e) indicate the significance of the analysis of variance.

**Table 4 foods-10-03039-t004:** Olfactory thresholds of 1,2-PG in water and its odor characteristics.

Compounds	Odor Threshold (in Pure Water) mg/L	Odor Characteristics
(*S*)-1,2-PG	23.92	Aromas of wood, faint alcoholic
(*R*)-1,2-PG	4.66	Sweet aroma, fruity, faint alcoholic

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
