# Peer review of "Distribution and Quantification of 1,2-Propylene Glycol Enantiomers in Baijiu"

_foods, 2021, doi:10.3390/foods10123039_

Round 1

Reviewer 1 Report

The manuscript entitled „ Distribution and quantification of 1,2-Propylene glycol enantiomers 2 in Baijiu” is of great interest and scientific value.

However, I have a few comments and questions.

  • In the tables I only see the standard deviation, the statistical analysis is missing.
  • The conclusion lacks information on the change in enantiomeric composition of 1,2-PG during the Baijiu ageing process.
  • Supplementary material (chromatograms) would enrich the article.

Overall, the manuscript needs a minor revision.

Reviewer 2 Report

In this study the authors investigated the enantiomeric distribution of 1,2-Propylene glycol (1,2-PG) in 64 commercial Chinese Baijiu including soy sauce aroma-type Baijiu (SSB), strong aroma-type Baijiu (STB), and light aroma-type Baijiu (LTB), via chiral gas chromatography (β-cyclodextrin) in order to establish a possible correlation of the enantiomeric ratio and aboundancies of 1,2-PG with aroma type and storage time.

The results obtained show that the content of 1,2-PG depends on the aroma style and age of the liquor to varying degrees, and its content is in the decreasing order of SSB, STB, and LTB. In particular, the enantiomeric ratios showed a predominance of the (S)-configuration in SSB, while the (R)-configuration dominated in STB. Furthermore the two configurations have different odors. From the obtained results the authors claim that the enantiomeric ratio of 1,2-PG could be a potential marker to differentiate the aroma styles of Baijiu. Moreover it might be one of the potential markers for adulteration control of Baijiu as industrial 1,2-PG is usually in the racemic mixture.

I think the paper is clear, relevant for the field and presented in a well-structured manner.

I Have only some very minor comments:

-please check interline 17/18; 42/43

-line 83 please check: “38 types of SSB, 16 of SSB

-line 182 please rephrase …”an average ratio of S/R of 87:13±3.17 in the SSB and about 89:11±1.15、89:11±3.82 in the vintage  ones, respectively. (Table 2).” For example: “…”an average ratio of S/R of 87:13±3.17 in the SSB while in the vintage ones about 89:11±1.15 and 89:11±3.82, respectively. (Table 2).”

-line 213 and 227 please add space after line 213 and after line 227

Line 236 please correct “an trend” with “a trend”

“Figure 3 b” the number 8 and 9 in the "aging" axis is somehow confusing since no sample of 8 or 9 year is reported, I suggest to remove them.

I think section “Data Availability Statement” and "Acknowledgment" could be removed since not filled.
